# Magnetic Compass Orientation in a Palaearctic–Indian Night Migrant, the Red-Headed Bunting

**DOI:** 10.3390/ani11061541

**Published:** 2021-05-25

**Authors:** Tushar Tyagi, Sanjay Kumar Bhardwaj

**Affiliations:** Department of Zoology, Ch. Charan Singh University, Meerut 250 004, Uttar Pradesh, India; tveertyagi@gmail.com

**Keywords:** earth’s magnetic field, magnetic compass, spring migration, orientation, temperature, Red-headed Buntings, *Emberiza bruniceps*

## Abstract

**Simple Summary:**

The earth’s magnetic field, celestial cues, and retention of geographical cues en route provide birds with compass knowledge during migration. The magnetic compass works on the direction of the magnetic field, specifically, the course of the field lines. We tested Red-headed Buntings in orientation cages in the evening during spring migration. Simulated overcast testing resulted in a northerly mean direction, while in clear skies, birds oriented in an NNW (north–northwest) direction. Buntings were exposed to 120° anticlockwise shifted magnetic fields under simulated overcast skies and responded by shifting their orientation accordingly. The results showed that this Palaearctic night migrant possesses a magnetic compass, as well as the fact that magnetic cues act as primary directional messengers. When birds were exposed to different environmental conditions at 22 °C and 38 °C temperatures under simulated overcast conditions, they showed a delay in *Zugunruhe* (migratory restlessness) at 22 °C, while an advance migratory restlessness was observed under 38 °C conditions. Hot and cold weather clearly influenced the timing of migrations in Red-headed Buntings, but not the direction.

**Abstract:**

Red-headed Buntings (*Emberiza bruniceps*) perform long-distance migrations within their southerly overwintering grounds and breeding areas in the northern hemisphere. Long-distance migration demands essential orientation mechanisms. The earth’s magnetic field, celestial cues, and memorization of geographical cues en route provide birds with compass knowledge during migration. Birds were tested during spring migration for orientation under natural clear skies, simulated overcast skies at natural day length and temperature, simulated overcast at 22 °C and 38 °C temperatures, and in the deflected (−120°) magnetic field. Under clear skies, the Red-headed Buntings were oriented NNW (north–northwest); simulated overcast testing resulted in a northerly mean direction at local temperatures as well as at 22 °C and 38 °C. The Buntings reacted strongly in favor of the rotated magnetic field under the simulated overcast sky, demonstrating the use of a magnetic compass for migrating in a specific direction.

## 1. Introduction

Billions of songbirds migrate from their wintering quarters to breeding quarters each year with their extraordinary navigation capabilities, covering several thousand kilometers following their endogenous circannual clock. Migratory birds take advantage of various compass systems so as to perform orientation throughout the migration, which includes information obtained from the earth’s magnetic field [1,2,3,4,5,6,7], star patterns [3,8,9,10,11,12,13,14], the sun’s position at dusk/polarized skylight patterns [15,16,17,18,19,20,21], and retention of geographical cues en route [22]. The strength of the geomagnetic field was suggested as a cue for avian navigation in the 19th century [23]. Birds can utilize the earth’s magnetic field for obtaining directional information during migration. Their magnetic inclination compass responds to the direction of the field lines and ignores the polarity of the geomagnetic field [5]. Close to the earth’s magnetic equator, the intensity of the geomagnetic field is decreased by one-half (less than 25,000 nT) in comparison with polar places (greater than 68,000 nT), and here, magnetic field lines are aligned along the parallel plane [24]. Birds during experiments have demonstrated acclimatization with decreased magnetic field strength, indicating that the magnetic compass was adjusted accordingly [2,25]. Translocated Eurasian Reed Warbler, *Acrocephalus scirpaceus*, a migratory songbird, could re-orient as a navigational response towards their migratory destinations using all the earth’s natural magnetic cues (magnetic field intensity, magnetic inclination, and magnetic declination) of unknown magnitude, but not when just one cue, i.e., magnetic declination, changed [26].

It has been suggested that animals may see the magnetic field details of a light-based radical-pair process that occurs in cryptochromes in the bird retina [6,27,28,29,30,31,32]. The brain region recognized as cluster N is needed for orientation among nocturnal migrating songbirds [33,34]. Long-distance nocturnal avian migrants normally go through 6 to 7 months in the wintering areas, 2 to 3 months in the breeding areas, and an additional 2 to 3 months in vernal and fall migration. Studies have shown that birds had already chosen the breeding grounds during the autumn prior to heading for migration, and therefore during the springtime, birds go to a place where they have already spent time and that is known to them [35,36]. Migratory orientation is not solely decided by the bird’s potential to use celestial cues or magnetic compass systems; however, it is additionally under direct control of environmental, sociological, and physiological factors [37]. Anticipating changes in the photoperiod is the fundamental cue utilized by numerous avian species to plan the transformation between the phases of the circannual cycle, including the use of additional cues, comprising temperature and food accessibility, to regulate the duration of the activities in the season [38]. The migration time in birds is also adjusted by responding to changes in temperature [39].

Birds often accumulate large fuel deposits on wintering grounds to help minimize the need for stopping times in the course of real migration. This could serve as a fundamental approach for migratory birds in order to reduce their vernal migration by adding more fuel during the premigratory period before they leave their wintering habitat. Birds may possess additional time so as to deposit premigratory fat during spring than in the autumn (whereas numerous bird species can be restricted by molt and reproductive requirements, and so forth, e.g., [40]). Following long-distance songbirds, namely, Wood Thrushes (*Hylocichla mustelina*) and Purple Martins (*Progne subis*), through applying geolocators, researchers demonstrated that the general migration rate was two to six times faster in springtime compared with autumn [41]. Most passerine night migrants were supposed to calibrate their directional preferences around sunset [4,16,21,42]. The early takeoffs post-sunset during vernal migration than autumn [43] could likewise turn into an impact of birds to show up before the expected time towards the breeding quarters by increasing their migration speed [44]. The information related to cues utilized by songbirds in the overwintering regions is scarce, particularly within the birds of the Indian subcontinent whose avian flocks are comparatively less researched across the globe [45].

The aim of the present study was to investigate whether migratory Red-headed Bunting orient in a specified direction in the course of migration under the following conditions:(a)Open skies when celestial cues (sunset, horizon glow, and skylight polarization patterns) were visible to the songbirds in the natural magnetic field.(b)Simulated overcast skies in the natural magnetic field with no celestial cues visible.(c)The magnetic field shifted −120° under simulated overcast skies.(d)The effect of different temperatures (22 °C and 38 °C) on orientation, tested under simulated overcast skies.

## 2. Materials and Methods

### 2.1. Study Species and Animal Husbandry

Red-headed Buntings (*Emberiza bruniceps*; Brandt 1841) are long-distance migrants that overwinter in India and leave to their breeding destinations in Central Asia (≈40° N) during spring migration in late March/early April [46]. The migration of Red-headed Buntings to the north begins in March, yet the main migratory season is in April and the beginning of May, showing up in breeding regions during the first half of May [47]. The orientation funnel experiments were carried out at C.C.S. University, Meerut (28°58’N, 77°44’E, India) in accordance with regulations laid by the Institutional Animal Ethics Committee (IAEC) under protocol number CCSU/IAEC/2018/meeting01/10 during the migratory season in spring 2018. Males often reach the reproducing ground before females [48]. Hence, adult male Red-headed Buntings were captured using mist nets from overwintering flock in January 2018 under the permit following the guidelines from Principal Chief Conservator of Forest, Uttar Pradesh, India and were kept in an outdoor aviary (dimensions = 3 m × 2.5 m × 2.5 m) in social groups under natural temperature and light conditions. Aviary was constructed mainly of aluminum mesh with round edges (to avoid foot injury), sturdy bars, and glass on a raised platform so that birds could be exposed to day and night sky in the local geomagnetic field. The aviary contains pesticide-free green shrubs, nest boxes, and wooden branches with varying diameters to allow perching, hopping, wing-whirring, and trimming of claws. Housing was constructed in such a way that it included a separate section for bathing and playing. Wood shavings in trays were kept on solid floors below the perches to collect fecal matter, and the flooring was cleaned daily with disinfectant to maintain sanitization. Birds were fed on foxtail millet (*Setaria italica*), shredded eggs, crumbled cheese, and vitaminized water ad libitum. Birds were monitored daily for health status. This will likewise enable familiarity with human faces and hands, which is probably going to diminish stress during handling and experimental procedure. Handling of birds was performed using disposable gloves, and a veterinarian was consulted in case of any signs of disease or abnormal behavior in birds. Birds were selected for orientation experiments when they started showing intense nighttime restlessness and had accumulated visible subcutaneous fat of ≥3. Fat deposits on various parts of the body (e.g., abdomen, furcular, and scapular) were analyzed using a score index 0–5. The value of 0 indicates no visible subcutaneous fat and 5 indicates a large fat deposit all over the body [49]. In total, we tested 44 birds from 14 April to 31 May 2018 (experiment 1, *n* = 22; experiment 2, *n* = 22) in orientation funnels. Birds were released in the fields near the campus in September 2018 when they attained overwintering physiology, and the health condition of birds was checked by a veterinarian before the final release.

### 2.2. Experimental Procedure

Birds were tested in the local earth’s magnetic field: total field intensity = 48,497 nT, inclination = 45.56°, mN (magnetic north) = 360°. Headings are provided with reference to mN; mN = geographic north 1.167°. Tests started 10 minutes before sunset [50] since the civil twilight period is of shorter duration (24–27 minutes) and continued for 60 minutes. Radar tracking studies have shown that passerine night migrants may leave closely before sunset, during the twilight interval, and at midnight [51]. Experiments were carried out in modified “Emlen funnels” made up of non-magnetic materials with a slope of 45°; height of 15.5 cm; and upper and lower diameters of 35 and 10 cm, respectively [52], to record the songbird’s migratory orientation. The lid of the orientation funnel was covered with a transparent plastic sheet, allowing the bird’s vision to approximately 160° of the sky above them. Simulated light and the moon did not have a significant impact on the orientation of birds, since the experiments were performed before absolute darkness [53]. The orientation of the birds was recorded with a DPRO-AD940 camera with 30 pieces of 940 nm IR LEDs that did not produce any glow. Data from video recordings of orienting birds was analyzed using BirdOriTrack software developed by Rachel Muheim (Version 7.11.0; MathWorks Inc., Natick, MA, USA; [54]). We tested the orientation of songbirds under the clear sky, where no landmarks were visible to birds, and under simulated overcast skies in rooms whose roofs were covered with non-transparent 3 mm plexiglass diffusing sheets. Experiments were conducted in two different conditions: (1) local earth’s magnetic field, and (2) magnetic north (mN) shifted 120° anticlockwise where total intensity and the inclination of the magnetic field remain unchanged. To make sure that the birds were habitual with the testing process and to minimize the possibility that the birds would behave differently when they were first introduced to the test environment, we tested each songbird at least thrice before the actual test.

### 2.3. Helmholtz Coil System

Helmholtz coil system (model HCS01CL, MEDA, Inc., Dulles, VA, USA) is a three-axial coil system used to maintain a uniform, constant, and accurate magnetic field at the center of the coil system by controlling the amount of independent current passing through each coil that produces the magnetic field in the direction parallel to its axis and perpendicular to other two coils using Fleming’s right-hand thumb rule. The region where the experiment took place was free from magnetic material as the ferromagnetic substance may affect the homogenous magnetic field of the coil system. HCS01CL rack assembly consists of high-power bipolar supplies (model BOP 72-6, Kepco, Flushing, NY, USA) that regulate current in *x*-, *y*-, and *z*-axes; CCM-3 magnetometer with fluxgate sensor; and the thermal control unit forms a negative feedback loop due to which field in the central region of the coil tends to zero and sensor produces the field equivalent to the total of the earth and the Helmholtz coil field is magnified by the magnetometer. Power supplies by BOPs enforce the Helmholtz coil to produce the field near fluxgate, which opposes the earth’s field; thus, a stable magnetic field will be maintained at the center of the coil. The computer has a PCB that is connected to CCM-3 that regulates the polarity and amount of current flowing through the coil under the administration of the operator who monitors the value of supply through the video display terminal (VDT) by changing the polarity and magnitude of the current through the keyboard. The experiment was conducted using a single power distribution unit (PDU) that supplies current to all the equipment.

### 2.4. Experiment 1

(a)Clear sky condition: Twenty-two birds, each tested four times in the natural magnetic field under the open sky. Sunset and horizon glow was clearly visible to the birds.(b)Simulated overcast condition: The same birds were tested under simulated overcast skies in rooms (dimension = 4 m × 4 m × 3 m) in the natural magnetic field. No celestial cues were visible to the birds. The simulated overcast sky was made using translucent plexiglass sheets of 3 mm thickness that allow natural light to pass through them.(c)−120° rotated magnetic field under simulated overcast condition: The same 22 birds were tested under simulated overcast skies in rooms (dimension = 4 m × 4 m × 3 m) in a 120° counter-clockwise rotated magnetic field with the help of the Helmholtz coil system. Under this experiment, no celestial cues were visible to the birds. Helmholtz coil apparatus HCS01CL of 2 m × 2 m × 2 m was used to rotate the horizontal component of the magnetic field. The experimental birds were exposed to a deflected magnetic field (−120°) indoors under the simulated overcast sky in orientation funnels for a period of 60 minutes.

### 2.5. Experiment 2

The experiments were conducted when temperatures in the outdoor aviaries were in the range of 22–38 °C during spring migration. The birds were tested for orientation under simulated overcast skies at 22 °C and 38 °C in the natural magnetic field: two wooden rooms (dimension = 4 m × 4 m × 3 m) were used for the experiments. Each room had a simulated overcast sky allowing only natural light to pass through them. Out of 22 birds, 11 birds were tested at 22 °C, while the other 11 were tested at 38 °C in separate rooms. Both groups of birds were interchanged regularly in two rooms every other day to remove bias. The temperature in rooms was maintained through a Quartz heater (model Orpat OQH-1230 230V 800W, Gujarat, India) along with a thermostat (model amiciSmart AC 220V 10A, Noida, UP, India) and a split air conditioner (model LG LSA5PW3A, LG Electronics India Pvt. Ltd., Noida, UP, India).

### 2.6. Statistics

Data from video recordings of test birds performing orientation were analyzed using BirdOriTrack software (MathWorks Inc., Natick, MA, USA). The Rayleigh-z test was used to evaluate the significance of the grand mean vector and examine the uniformity of the distribution at a 95% confidence interval. Analysis of circular data was conducted using Oriana (v4.02, Kovach Computing Services, Pentraeth, Wales, UK). First-order and second-order statistics were used to determine the mean heading of Red-headed Buntings. Homogeneity in the directional mean was evaluated through the parametric Watson–Williams F-test, assuming that the sample was obtained from von Mises distribution with mean vector length, (r) ≥ 0.75. GraphPad Prism version 9.0.0.121 (San Diego, CA, USA) was used to perform Student’s *t*-test to compare the number of hops of birds at different temperatures. We discarded bird-hours that showed less than 40 hops, or those that were unsuccessful in performing a sensibly well-specified orientation [55].

## 3. Results

### 3.1. Orientation in the Local Magnetic Field

When birds were exposed to clear skies in the Emlen funnel, we found that sunset, horizon glow, and natural skylight polarization patterns were clearly visible. The direction of mean headings was appropriate for spring migration (341° ± 12° at a confidence interval of 95%, r = 0.872, *p* < 0.0001, *n* = 22; Figure 1a), which was significantly different from the mean sun azimuth at the middle of the test hour (288° ± 1° at a confidence interval of 95%). The phototactic or menotactic effect was not noticeable during clear sky experiments.

The same birds were tested under the simulated overcast sky in rooms with no availability of celestial cues in the natural magnetic field oriented highly significantly towards the north (358° ± 18° at a confidence interval of 95%, r = 0.752, *p* < 0.0001, *n* = 22; Figure 1b). These headings were not significantly different from the headings of the birds tested under clear skies (95% confidence interval overlap; Watson–Williams F-test: F = 2.342, *p* = 0.133).

### 3.2. Orientation in a −120° Rotated Magnetic Field under Simulated Overcast Skies

The same birds were tested under the simulated overcast sky in a 120° counter-clockwise rotated magnetic field, which was responded to by shifting their orientation accordingly in the expected direction. The mean orientation of birds in the changed magnetic field (236° ± 16°, r = 0.79, *p* < 0.0001, *n* = 22; Figure 1c) differed significantly and in the expected direction from the same birds’ orientation under simulated overcast skies in the local natural magnetic field (95% confidence interval did not overlap; Watson–Williams F-test: F = 83.17, *p* < 0.0001).

### 3.3. Orientation of Birds Tested at 22 °C and 38 °C in the Natural Magnetic Field under Simulated Overcast Skies

The mean orientation of birds at 22 °C (350° ± 18°, r = 0.758, *p* < 0.0001, *n* = 22; Figure 2a) did not differ significantly from the same birds tested at 38 °C (356° ± 15°, r = 0.814, *p* < 0.0001, *n* = 22; Figure 2b), and the 95% confidence interval coincided (Watson–Williams F-test: F = 0.269, *p* = 0.607). Birds tested at 22 °C under the simulated overcast sky showed fewer hops as compared to birds at 38 °C in orientation funnels (Student’s *t*-test: *t* = 6.599, dF = 42, *p* < 0.0001).

## 4. Discussion

Most investigations into the potential use of the earth’s magnetic field as a directional reference for bird migratory orientation focused on passerine birds as experimental subjects. We discovered no significant difference between the mean orientation of the Red-headed Buntings tested under clear and simulated overcast sky conditions, demonstrating similar directional preference. Red-headed Buntings tested under simulated overcast conditions displayed an intense migratory activity similar to birds tested under clear skies and showed a significant northerly mean orientation, illustrating that they do not have any trouble in orienting themselves under simulated overcast skies. When birds were tested in an open field under the clear sky, they showed NNW (Figure 1a) orientation. It is possible that birds were able to adapt to the visible cues and could orient themselves according to sunset/polarized light pattern. Under simulated overcast conditions, adult male Red-headed Buntings with migratory experience oriented themselves with their magnetic compass rather than the setting sun stimuli. No celestial cues were visible to birds in the indoor facility. The results indicate that Buntings can also choose seasonally relevant migratory directions in the absence of sunset/polarized patterns. These results are in agreement with the discovery of an earlier study on European Robins, *Erithacus rubecula*, a night migrant [56]. The results of the orientation experiments under clear versus overcast skies conducted on a flat rooftop during spring migration with Palaearctic nocturnal migrant European Robins, *Erithacus rubecula*, demonstrated that birds oriented significantly under both clear and overcast skies in a seasonally meaningful migratory direction [57]. The orientation funnel experiments clearly demonstrate the ability of birds to perform orientation without an approach to celestial cues [58,59,60,61,62,63]. They appear to have the ability to determine a compass route utilizing magnetic cues. On the basis of radar studies, research has observed that nearly one-third of night departures took place in weather conditions usually thought unsuitable for migration. A study in southern Louisiana showed that with the exception of heavy rainfall, dense overcast skies have not delayed night migration, nor have they reduced the number of migrants [64]. Experiments with different passerine birds have shown that magnetic guidance alone acts as a reference for migratory orientation [26,65,66,67,68,69,70,71,72]. When magnetic north was turned to 120° counterclockwise under simulated overcast skies, songbirds closely followed a −120° shift and oriented themselves accordingly. These findings indicate that a functional magnetic compass exists in migratory Red-headed Buntings. Earlier studies on European Robins showed that in closed rooms they pursue a change in the mN direction instantly [73,74]. Orientation experiments under the rotated magnetic field provided the orientation tendency in accordance with the magnetic field shifts [75,76,77,78,79].

The majority of the birds tested during this investigation were oriented in what should be considered to be seasonally relevant directions. It is possible that the dissimilarity noticed in the orientation between spring migratory birds indicates different migration objectives. The fact that the total number of birds were caught from the same overwintering community in no way means that they live in the same breeding ground [80,81]. The investigations introduced at this point show that Red-headed Buntings are competent in identifying and utilizing the earth’s magnetic field for their migratory orientation. The similarity of the outcomes acquired with and without the access to celestial cues during vernal migration implies that the Red-headed Buntings depend fundamentally on their magnetic compass for orientation close to sunset. From that perspective, our outcomes are equivalent to the discoveries stated in the experiments performed on Snow Buntings during spring migration. The Snow Buntings, *Plectrophenax nivalis*, when tested under clear and simulated overcast conditions at sunset, showed that the geomagnetic field was the essential orientation reference. The Snow Buntings strictly followed an anticlockwise rotation of the magnetic north [55]. Corresponding results were obtained in outdoor experiments with Dunnocks, *Prunella modularis*, under clear skies around sunset, finding that the birds followed a clockwise deflection of magnetic north [79]. Experiments conducted close to the magnetic North Pole at steep inclination angles of geomagnetic field have revealed that White-crowned Sparrows, *Zonotrichia leucophrys gambelii*, have the ability to use inherited magnetic compass for orientation in a seasonally meaningful direction tested under clear and simulated overcast skies during autumn migration [62]. The magnetic compass would have a simpler application with growing distance from the magnetic north pole, as the earth’s magnetic field would not be so steep but instead rather uniform (less deviation in declination [24]).

Most of the Red-headed Buntings exposed to 22 °C and 38 °C in the local earth’s magnetic field oriented in a seasonally appropriate northerly direction. The mean directional preferences of the birds tested at 22 °C did not deviate significantly from the birds tested at 38 °C. Radar observations performed on passerines in southern England from 1960 to 1963 showed that there was an increased spring migration density with a rise in temperature by about 4 °C at 18:00 hours [82]. The pattern revealed in earlier investigations showed that the Red-breasted Flycatcher, *Ficedula parva*, had been showing up before, and the approaching time was impacted by temperature. The authors indicated that appearance times in Poland depend on regional temperature as well as on temperature during the migratory course and in the wintering region of the Indian subcontinent [45]. We estimated the migratory restlessness during the evening to possibly derive how birds who choose to leave on a nighttime migratory session may be affected by climatic conditions. Our outcomes are uniform with slow-down in a northerly orientation during spring migration against upcoming cold weather and are similar to outcomes demonstrated in free-living Thrushes [39]. Our results were similar to White-throated Sparrows, *Zonotrichia albicollis*, when presented to temperature/pressure changes. The drop in temperature led to a reduction in migratory restlessness during the early hours of the night; however, there was no further impact of pressure changes [83]. Red-headed Buntings react to changes in air temperature; at the point when birds were in the vernal migratory stage, we found that birds tested at 22 °C decreased the number of valid hops in comparison to birds tested at 38 °C. However, further trials are needed to discuss the impacts of varying temperatures alone. 

Our findings are in accordance with former observations performed on a long-distance migrant that demonstrates that increased temperatures expanded the likelihood of take-off in the Pink-footed Goose, *Anser brachyrhynchus*, during spring migration [84]. During springtime, if the temperature fall is significant, the birds should slow down the migration or even opposite migration. Estimates show that both continuous flight and stopover are expensive because of higher energy costs in cooler weather [85]. It may be deduced from earlier studies that the continuation of spring migration in Pied Flycatchers (*Ficedula hypoleuca*) is largely governed by temperature along the way [86]. Our results are in concurrence with earlier studies that had shown the rate of migration inside North America increments with temperature [87]. Previous investigations indicate that the Pied Flycatchers can accelerate their migration because of increasing temperatures during the course of spring migration [88]. Temperature is stated to influence the procedure and time of avian breeding, either straightforwardly by changing activity pattern and performance, or indirectly by interfacing with photoperiod and altering its impact [89]. Accordingly, high temperature (38 °C) quickened the pace of testicular response than at low temperature (22 °C) after long photoperiodic stimulation that induced the migratory restlessness in Red-headed Buntings [90]. In our knowledge, this is the first account of the migratory orientation behavior of an Indian songbird. 

## 5. Conclusions

The results depict that this Palaearctic night migrant possesses a magnetic compass, and magnetic cues act as primary directional messengers. Temperature plays a crucial role in controlling the timing of migration besides the photoperiod. Most studies on the migratory orientation behavior of songbirds have been performed in the laboratory; however, many more studies still need to be performed on songbirds, especially of the Indian subcontinent, using “light-level geolocation data loggers”, geologgers that can gain in-depth knowledge of their migratory routes [91].

## Figures and Tables

**Figure 1 animals-11-01541-f001:**
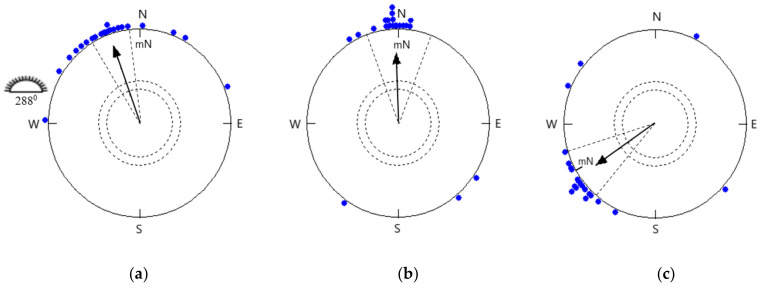
Directional preference of Red-headed Buntings at the experimental site under (**a**) clear sky (α = 341°, *n* = 22, r = 0.872, *p* < 0.0001; 95% CI = 329°–353°), (**b**) simulated overcast sky (α = 358°, *n* = 22, r = 0.752, *p* < 0.0001; 95% CI = 340°–16°), and (**c**) 120° counter-clockwise rotated magnetic field under simulated overcast skies (α = 236°, *n* = 22, r = 0.79, *p* < 0.0001; 95% CI = 220°–252°) during spring migration. The solid arrow represents the direction (α) and length (r) of the grand mean vector. Each symbol (filled dot) on the circumference of the circle depicts the mean heading of an individual bird tested 4 times. The inner and outer dashed circles represent the minimal length of mean vector r at 95% (*p* < 0.05) and 99% (*p* < 0.01) significance according to the Rayleigh test; radial dashed lines represent the confidence interval (CI), mN denotes the direction of the magnetic north, and the sun symbolizes the mean azimuth angle calculated in the halfway of the experimental hour.

**Figure 2 animals-11-01541-f002:**
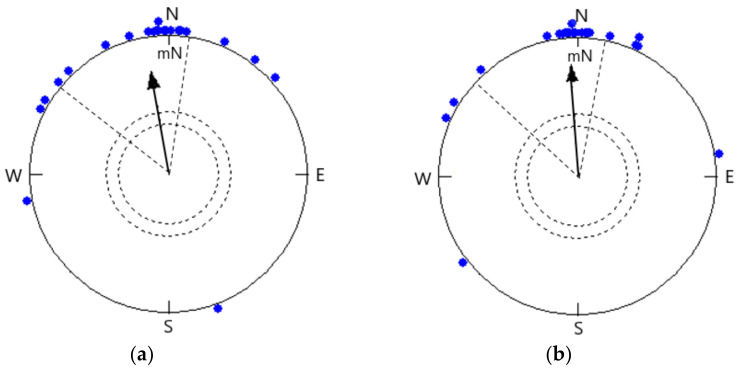
Orientation of Red-headed Buntings under simulated overcast conditions at (**a**) 22 °C temperature (α = 350°, *n* = 22, r = 0.758, *p* < 0.0001; 95% CI = 332°–8°) and (**b**) 38 °C temperature (α = 356°, *n* = 22 (the same birds were tested at 22 °C), r = 0.814, *p* < 0.0001; 95% CI = 341°–11°). Each symbol (filled dot) on the circumference of the circle depicts the mean heading of an individual bird tested 3 times.

## Data Availability

The dataset used and/or analyzed during the current study is available from the corresponding author upon reasonable request.

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
