# Peer review of "Magnetic Compass Orientation in a Palaearctic–Indian Night Migrant, the Red-Headed Bunting"

_animals, 2021, doi:10.3390/ani11061541_

Round 1

Reviewer 1 Report

A useful and interesting paper on a key behaviour whereby expanding knowledge is applicable across a wide range of species. I would be happy for this paper to be published if more information on the ethics of the research were provided. Presently, not enough information is provided on the ethics of the this research and therefore I cannot recommend for publication. 

Ethics. Please provide more information on the licensing and permits used to catch the birds. If no permits were obtained then I cannot recommend this paper for publication on ethical grounds.

How were birds handled? How long were they housed for? Where they kept in social groups? Do you have permissions to catch and hold wild animals? Please provide more information on the husbandry and management of free living animals used for research.

You say that you scored for fat deposits on the birds, and that some individuals could be love in fat. Did you still use these birds in your studies because they should have been released or left to recover. This links back to the need for more explanation of the use of wild birds and the ethical applications provided. 

Check the grammar of the common name, I believe it should be red-headed bunting but you switch and change between a hyphen and no hyphen, capitals and no capitals. 

Please get a thorough proof read for written English, to check on basic spelling and grammar. Perhaps reach out to a native speaker? 

You say that birds underwent multiple testing. Please explain how birds were rested and how did you know they were habituated? What if they were chronically stressed and this was learned helplessness rather than habituation? 

Were birds prepared for release back into the field? How do you know that fitness was not compromised by captive housing? Again, more information on the length of time that birds were housed for would allow this to be judged. 

You need to provide more information on how statistical testing was implemented. What tests were used for what specific data and why? Explain your testing in line with each hypothesis. 

Please revisit your paper, and rewrite your methods to consider all of the points I raise above concerning ethics and animal welfare, the housing and husbandry of the birds and how they were used in testing. How long they were kept for and when they were released, how they were released and where they were released. Who was involved with catching birds and who did the handling and restraint?

Reviewer 2 Report

See attached file

Author Response

Thank you for providing your crucial feedback. We have implemented the changes in our text as per your suggestion and removed the misunderstanding as indicated by you in simple summary 2nd sentence and line 38-39. We also provided the angles in full degree as per your suggestion and we truly appreciate them. We also removed decimal from nanotesla in intensity.

Thank you so much for providing your insightful review on our article.

Reviewer 3 Report

This is a very well-performed study on the magnetic compass orientation of an Indian Subcontinent, night-migratory songbird. Almost all previous knowledge about the navigation mechanisms of birds rely on data from European, North American, and a few Australian birds. This, I think, is the first study of its kind conducted in India. The study is very clearly designed. The equipment and methodologies used are excellent, and the birds are orienting exceptionally well using their magnetic compass only and the very clearly follow a horizontal rotation of the magnetic field. While this study will not revolutionize the field, it is a very clean and well-done study in a part of the world where we had no data previously. The directness of the magnetic  orientation of these redheaded buntings is among the best by any species tested anywhere in the world so far, which is very nice. The test of the birds orientation under different temperatures is a nice addition even though temperature - as I would have expected - did not affect their directionality.

The text is very readable and fully understandable even though there are some cases where strange English terms are used that I am sure a native English speaker would have corrected. The paper would benefit from linguistic editing by a native English speaker - but it would be readable and fully understandable even without it.

Here comes a few minor suggestions: 

Line 41: Maybe you also want to refer to the most recent review on the topic: 

Mouritsen, H. (2018) Long-distance navigation and magnetoreception in migratory animals. Nature 558, 50-59. doi: 10.1038/s41586-018-0176-1   Line 51: "recommending" should be "indicating". Line 57: "recommended" should be "suggested". Line 81: "more fast" should be "faster". Line 104: "principle migratory course" should be "main migratory season" Line: 147: Which company produced the coils? Line 154: What is the company and model of the used power supplies? Kepco? BOP 50-2? Line 231: "lower" should be "fewer". Line 266: "uniform" should be "in agreement". Line 273: "There appears to have the option ..." should be "They appear to have the ability ..." Line 293: "approach" should be "access".  

Author Response

Thank you for providing your crucial feedback. We have added the latest reference in line 41 “Mouritsen, H. (2018) Long-distance navigation and magnetoreception in migratory animals. Nature 558, 50-59. doi: 10.1038/s41586-018-0176-1” as per your suggestion and its relevance with our text.

We also corrected some minor changes (in line 51, 57, 81, 104, 235, 269, 276, 296) as indicated by you and we truly appreciate them.

 "Line: 151: Which company produced the coils?” In the text we added MEDA, Inc company to make it more clear regarding the company that produced the Helmholtz coils system.

“Line 158: What is the company and model of the used power supplies?” We also added Kepco and the model BOP 72-6 as indicated by you to add more specific details.

Thank you so much for providing your insightful review on our article.

Round 2

Reviewer 1 Report

A much improved and clear paper with ethical considerations explained and detailed.

Before I recommend this for publication, whilst I appreciate that the authors have explained the ethical permits for the project, it is still not enough to say that "the birds were cared for properly". The husbandry and housing of the buntings, amount of time handled, how they were released, and how welfare was maintained, needs to be explained fully in the methods. Otherwise the paper's experimental design is not repeatable. Please provide a paragraph on animal husbandry. 

Author Response

Thank you for providing your crucial feedback. We have updated our text and added more information regarding housing conditions and added an additional paragraph on animal husbandry line 112-124. We have added information about the round edges in aviary that prevent foot injury and explained about the separation of section for bathing and playing. We also updated information about how sanitization was maintained in aviary. We explained the housing that includes pesticide-free green shrubs, wooden branches with varying diameters were present in aviary which allowed perching, hopping, wing-whirring and trimming of claws. We also explained the usage of wooden shavings to collect faecal matter which was cleaned daily to maintain sanitization and welfare of birds. We also updated the release information of birds in line 130-132. Hopefully, we satisfy all the requirements.  

Thank you once again for providing your inestimable suggestions.